# Toward the Integration of Technology-Based Interventions in the Care Pathway for People with Dementia: A Cross-National Study

**DOI:** 10.3390/ijerph181910405

**Published:** 2021-10-02

**Authors:** Vera Stara, Benjamin Vera, Daniel Bolliger, Susy Paolini, Michiel de Jong, Elisa Felici, Stephanie Koenderink, Lorena Rossi, Viviane Von Doellen, Mirko di Rosa

**Affiliations:** 1Models for Ageing Care and Technology, IRCCS INRCA-National Institute of Health and Science on Aging, 60124 Ancona, Italy; e.felici@inrca.it (E.F.); l.rossi@inrca.it (L.R.); 2iHome Lab, University of Applied Sciences & Arts, 6048 Lucerne, Switzerland; benjamin.vera@hslu.ch (B.V.); daniel.bolliger@hslu.ch (D.B.); 3Unit of Neurology, IRCCS INRCA-National Institute of Health and Science on Aging, 60124 Ancona, Italy; s.paolini@inrca.it; 4Research Group IT Innovations in Healthcare, Windesheim University of Applied Sciences, 8017 Zwolle, The Netherlands; M.de.jong@windesheim.nl; 5De Parabol, 7433 Schalkhaar, The Netherlands; S.Koenderink@deparabool.nl; 6One More Project OMP SàRL-S, 7570 Mersch, Luxembourg; vivianevondoellen@gmail.com; 7Unit of Geriatric Pharmacoepidemiology and Biostatistics, IRCCS INRCA-National Institute of Health and Science on Aging, 60124 Ancona, Italy; M.DIROSA@inrca.it

**Keywords:** older adults with dementia, embodied conversational agent, virtual agent, user-centered design, technology-based intervention, personas, cluster analysis, care pathway

## Abstract

Background: The integration of technology-based interventions into health and care provision in our aging society is still a challenge especially in the care pathway for people with dementia. Objective: The study aims to: (1) identify which socio-demographic characteristics are independently associated with the use of the embodied conversational agent among subjects with dementia, (2) uncover patient cluster profiles based on these characteristics, and (3) discuss technology-based interventions challenges. Methods: A virtual agent was used for four weeks by 55 persons with dementia living in their home environment. Results: Participants evaluated the agent as easy-to-use and quickly learnable. They felt confident while using the system and expressed the willingness to use it frequently. Moreover, 21/55 of the patients perceived the virtual agent as a friend and assistant who they could feel close to and who would remind them of important things. Conclusions: Technology-based interventions require a significant effort, such as personalized features and patient-centered care pathways, to be effective. Therefore, this study enriches the open discussion on how such virtual agents must be evidence-based related and designed by multidisciplinary teams, following patient-centered care as well as user-centered design approaches.

## 1. Introduction

The challenges from demographic change and aging are pushing the innovation of digital solutions to transform health and care provision worldwide.

This process started two decades ago until the onset of COVID-19 supplied an unprecedented stimulus [1,2]. However, there is still a need to identify key barriers and enablers to understand the challenges and the factors that could support the deployment of digital technologies in meeting the need for care and support of the aging population. This need is of particular interest in frail and vulnerable individuals, such as people living with dementia. It will affect around 47 to 132 million people by 2050, causing high impacts on individuals, families, communities, governments, and societies worldwide [3,4].

In this scenario, non-pharmacological interventions might play a vital role to improve quality of life and independence. Indeed, a broad spectrum of Information and Communication Technologies (ICTs) are seen as a way to compensate for cognitive decline and promote self-management to remain independent for as long as possible [5,6,7]. Significant optimism has recently emerged with the use of screen-based entities designed to encourage human face-to-face conversation skills [6]. Such virtual entities, called embodied conversational agents (ECAs) or personal virtual assistants (PVAs) [8,9,10,11,12,13,14,15], are natural, personalized, and human-like 2D and 3D chatbots or wholly embodied conversational agents acting in ambient-assisted living environments [16,17]. Due to this human resemblance, this technology reached the attention of the health sector, especially in the area of health counseling, coaching, and self-monitoring [18].

Despite this significant and growing interest, integrating such innovations into health and care provision in our aging society is still a challenge. The reasons are based on the lack of solid evidence that technology-based interventions can deliver care benefits to people with dementia. This lack of evidence is due to the substantial difficulties in researching within this field. Indeed, involving individuals with dementia, and the possibility of engaging them in using new technologies for a significant period, is a common limitation that impacts the generalization of results [8]. Moreover, these individuals have often low digital experience and decreasing abilities in managing routine changes (i.e., introducing a new device into their daily life) along the path defined by the progression of symptoms. Therefore, becoming familiar with the technology at an early stage of dementia is fundamental to design an effective technology-based intervention. This latter point cannot be achieved without a strong and effective health care plan that promotes the adoption of new technologies. Especially during the COVID-19 pandemic, the need to increase digital access to health information and the general need to digitally communicate with patients and care receivers became evident. This then led to increased use of existing ICT solutions and stimulated the development of such innovations. Indeed, artificial intelligence brings the benefit of 24 × 7 health monitoring. Based on optimization algorithms such technologies continually collect and analyze data in order to assess the risk level and share behavioral or care recommendations. Furthermore, this remote digital assistance can increase, not replace, the ability of healthcare professionals to better understand the day-to-day patterns and needs of the people they care for, thus facilitating the quality of life of people with dementia and their caregivers. Despite this unprecedented stimulus, there is still an urgent need to discuss how digital technologies support the delivery of services and care, especially in vulnerable people with limited digital literacy skills.

The study aims to: (1) identify which socio-demographic characteristics are independently associated with the use of the embodied conversational agent called Anne, among subjects with dementia after four weeks of usage in their home environments, (2) uncover patient cluster profiles based on these characteristics, and (3) discuss how technology-based interventions require a significant effort such as personalized features and patient-centered care pathways, to be effective for the different types of users.

## 2. Materials and Methods

### 2.1. Anne, the Embodied Conversational Agent

This study used data collected during the European-funded project Living Well With Anne. The ECA was adapted for people living with forgetfulness, as is typical at the beginning of dementia. All the functionalities and features of the system shown in Figure 1, were developed following a user-driven approach, with the engagement of a multidisciplinary team and the involvement of users in the requirements definition process [19]. The virtual assistant works on a Surface Pro tablet under the Microsoft Windows 10 operating system and supports Dutch, English, German, Italian, and French. Anne was designed to help people with dementia in their daily routine: communication with the outside world (Call), keeping track of items on the personal calendar and daily structure (Agenda), medication plan adherence (Medications), reading the news (News) and relaxation (Games, Album and Radio). The interaction modality is based on two possible channels: (1) a visual and haptic channel, via a graphical user interface where the user can look at the screen and touch, and (2) an acoustic channel, via a voice user interface where the user can listen to the avatar’s voice and speak to it. Anne’s voice user interface consists of automated speech recognition and text-to-speech functions.

In comparison with other ECAs designed for people with dementia, this virtual agent represents a step forward: (1) Anne is based on a mobile standalone solution, whereas the other examples of virtual agents are displayed on standard television sets and computer screens; (2) Anne offers a multipurpose tool integrating features such as reminders (personal and medication agenda), communication (video calls), information (news), and entertainment (games and music) that support users in all aspects of daily life engaging them in various activities (i.e., help them pass the time in a more meaningful way and improve their quality of life). On the contrary, other ECA functions sought mainly to overcome memory problems, guide patients in their daily activities, and meet their need for communication and social interaction [8].

### 2.2. Subjects

Based on the requirements analysis, the clinical staff involved in the study suggested a suitable target group for Anne (Box 1): active and independent users able to interact with the avatar, discuss their thoughts, and express their opinions about it and who are competent and able to answer the protocol used for gathering data before and after the trial. For safety reasons and, to avoid any possible unpredictable dramatic swings in mood and behavior in front of an unknown virtual assistant, individuals in the advanced stages of dementia were excluded from this study (Box 2). Fifty-five volunteers diagnosed with dementia were enrolled in the study in three different sites: 20 users in Italy, 20 users in Luxembourg, and 15 users in The Netherlands.

Box 1Inclusion criteria for the study.
**Inclusion Criteria**

Age of 60 years or olderLiving independentlyMini-Mental Status Examination (MMSE) [20] score between 24 and 30Ability to understand and sign the written informed consent


Box 2Exclusion criteria for the study: the presence of at least one of the following criteria excluded the user from enrolment.
**Exclusion Criteria**

Lack of written informed consentPresence of an unstable chronic condition, with a Mini-Mental Status Examination (MMSE) score <24Presence of severe physical illness or disabilities that could be aggravated through the use of Anne


### 2.3. Recruitment Procedure

The enrolment and recruitment strategy was performed in the city of Ancona (Italy) at the Neurology Unit of the IRCCS INRCA-National Institute of Health and Science on Aging, in Luxembourg by the community care provider Stëftung Hëllef Doheem(SHD) and in the town of Deventer at de Parabool Care Center. According to the Living Well with Anne project activities, each site identified 20 to 25 eligible participants among those with regular access to the centers or being visited regularly by professional caregivers. Each of them was invited to test the virtual agent Anne. Six persons refused to participate. The 55 users (20 in Italy, 20 in Luxembourg, and 15 in The Netherlands) who voluntarily accept participating in the research were enrolled. After the first week of enrolment, four participants dropped out of the investigation due to a lack of interest in using the system. They were substituted with the other four volunteers on the reserve list. Each local ethical committee approved this study, and all subjects provided their informed written consent. Data collection was performed during January and February 2020.

### 2.4. Study Design

Similar to Stara et al. [19], a mixed-method study design was used to gather data from users. Data collected was then used to uncover individuals’ cluster characteristics and designed personas accordingly. At first, each enrolled subject was introduced to Anne, receiving general training on its correct use, and given a printed user manual with step-by-step instructions and a dedicated phone number to call in case of technical problems or doubts. Users were asked to interact with Anne for four weeks in their homes. Each instrument was verbally administered in face-to-face sessions by a trained caregiving staff such as psychologists, who entered the response on a paper version.

To assess the health status of participants, users responded to the following test at the beginning and end of the four weeks of usage:Quality of life in older adults with cognitive impairment (Quality of Life in Alzheimer Disease scale [QOL-AD]) questionnaire [21,22]. It is composed of 13 items covering physical health, energy, mood, living situations, memory, family, marriage, friends, chores, fun, money, self, and life as a whole. The assessment is scored on a 4-point Likert scale ranging from 1 (poor) to 4 (excellent), with total scores ranging from 13 to 52.At the end of the period, users also responded to the questionnaires below:The System Usability Scale (SUS) [23] provides a quantitative measure of the usability of a system. It is composed of ten statements rated by a 5-point Likert scale scored from 0–100, with 100 indicating perfect usability. This score is usually compared and interpreted considering the acceptable average value of 68 (SD 12.5), which was determined for a variety of products and tools, including websites and technologies, provided by Sauro and Lewis [24] after the analysis of more than 5000 user scores encompassing almost 500 studies.The closeness scale [25] assesses the perceived relationship by asking respondents to evaluate their relationship with Anne. Respondents had to select 1 of 7 pairs of increasingly overlapping circles that best described their relationship with Anne. In each pair of circles, one circle referred to the respondent, and the other circle referred to Anne. A larger overlap indicated a closer relationship. For the analysis, visualization was numbered as follows: 1 = no overlap, 2 = little overlap, 3 = some overlap, 4 = equal overlap, 5 = strong overlap, 6 = very strong overlap, and 7 = almost total overlap.Some unstructured short questions were asked to users in order to record the role attribute to Anne.

Moreover, during the four weeks of use, telemetry data was collected to track every event caused by the user’s activity on the tablet. Telemetry is the process of collecting data about remote objects and sending it to a computer electronically. These activities include clicks on the touch screen or voice interaction. Furthermore, the used feature types, such as games or medication reminders, were also recorded. All these activities were timestamped and therefore enabled a comprehensive analysis of the user’s behavior throughout time. These usage records are appropriate to detect problems and evaluate the status quo [26].

Two distinct classes of events were established in order to analyze telemetry data meaningfully. Transition events describe events for navigating through Anne, which are mostly the events caused by touching on the device’s touchscreen and, target events which include the usage of Anne’s actual features such as reading the news and listening to the radio. Due to the similarity with the mouse navigation, a touch on the device’s touchscreen is called a click in the sequel. Let us illustrate target and transition events in a realistic example: A user would like to listen to the radio. As he just started to use Anne, he navigates by mistake to the news menu, realizes that, and then navigates back to the main menu. Afterwards, he navigates to the radio menu and clicks on a radio channel that he intended to listen to. Each navigation caused a transition event as well as a click on the radio channel. This results in 4 transition events and one target event. The number of transition events to reach a target event depends on the target event itself. Due to the user-friendliness of Anne, a target event usually only requires 1 to 2 transition events. From this example, we also see how the distinction of transition and target events helps us assess Anne’s usability.

### 2.5. Statistical Analysis

In the descriptive analysis, continuous variables were reported as the mean and standard deviation, while categorical variables were expressed as absolute numbers and percentages for the whole sample and by country. An agglomerative hierarchical clustering was applied out in order to combine subjects with similar characteristics in the same group. Subjects’ characteristics considered for the cluster analysis were: age, gender, education, MMSE score, and previous experience with a tablet. A cluster fusion (Ward) procedure was carried out: such agglomerative procedures first combined all same characteristic combinations to a cluster. As soon as identical subjects’ typologies could not be combined anymore, two clusters were fused by, which means internal heterogeneity was least increased. This process was performed until the last fusion step when the last two remaining clusters were merged into one. From this point, the individual fusion steps were followed backwards in order to determine with the inverse screen test at which step the heterogeneity increased erratically (Elbow criterion). The optimal number of clusters was assessed from the analysis of the dendrogram.

## 3. Results

As shown in Table 1, the sample comprised 55 users (mean age 70.8 years, SD 9.9) where 36% were male and 64% were female. The general quality of life was in between the fair and good perception and maintained this level during the 4 weeks of the study. The majority of participants were married (47%) or single (30%) with a medium or high level of education. Twenty-two participants had previous experience using tablets whereas 26 did not report any use.

### 3.1. Telemetry Data and Usability

During the observed trial period, country usage was different. This can be more specifically illustrated by considering the median of target resp. transition events of a user per country:Italy: 417 target events and 1.334 transition eventsLuxembourg: 252 target events and 984 transition eventsNetherlands: 90 target events and 572 transition events

More intuitively, this means, for example, that a user living in Italy has typically 417 target events over the specified usage period. The Netherlands is usually the least active country, and Luxembourg is in-between. The only exception is the number of active days where Luxembourg is the least active (17 days on median) and Netherlands (26 days on median) is in-between. Italy is still the most active country, with 27 days on the median. The activity ranking as Italy is the most active followed by Luxembourg, and then the Netherlands is frequently found. Moreover, Italy has about twice as much as an activity than Luxembourg. Compared to Luxembourg, the Netherlands often has half as much activity. The box plot in Figure 2 illustrates it intuitively. In this figure, a large rank corresponds to a large number of target events of a user (on median).

The data showed that users often tend to use the same functionalities of Anne over the day. In order to identify the daily patterns, we divide a day into the following sections:Early morning: From 6 to 8 o’clockMorning: From 9 to 10 o’clockLunchtime: From 11 to 13 o’clockAfternoon: From 14 to 17 o’clockEvening: From 18 to 22 o’clockNight: From 23 to 5 o’clock

We focus here on target events as they represent the actual user’s intention. In the Early Morning, Morning, Evening, and Night medication events strongly dominate compared to others. These events are based on the confirmation of taking his medications.

We see that during the day sections where meals are consumed, users usually confirm to take their medications that are often medically prescribed to be taken during mealtimes. This gives a plausible explanation of this pattern. During Lunchtime and Afternoon, Anne is usually used for entertainment, particularly for playing games, listening to the radio, and reading the news. One of Anne’s most interesting functionalities was that users could also handle Anne by voice. The voice commands included reading news articles, showing an album as well as medication, telling the time, and initiating a call. Voice commands were efficient because with one command the target event can be initiated directly. However, it should be noted that not all events can be triggered by voice; especially, the very frequent game events that only work with a touchscreen. Thus, comparing touchscreen and voice interaction directly was difficult. Overall, 111,684 target events were initiated by using the touchscreen compared to 2545 using voice interaction. The preferred target event triggered by voice was news (699 cases) followed by medication (312 cases). In the case of the touchscreen, it was game (99,080 cases) followed by medication (6781 cases). Luxembourg was led in terms of successful speech interactions (18 on median per user) closely followed by Italy (16 on median per user) and then with more distance Netherlands (8 on median per user). Anne’s medication, news, and game features were very frequently used in Italy and Luxembourg and rather rarely in the Netherlands. An exceptional feature that did not obey the activity ranking is radio. In the case of the radio feature, Italy was still the most active (36 events on median), followed by the Netherlands (28 events on median). However, Luxembourg did not practically use it for technical problems due to encodings (0 events on median).

Figure 3 shows the number of users over time in the different countries. In any country, the decrease of users is strongest during the first period (1–6 days) and usually flattens afterwards. In the first period, users seem very likely to drop out, because of unfamiliarity with Anne. In the subsequent periods, the users professionalize their usage and integrate it into their everyday life. Italy and the Netherlands had a very similar trend. However, Luxembourg data evidenced a strong decrease in the periods 1–6 days and 21–27 days.

The plots in Figure 4 show the number of target and transition events over time per country. Per definition, transition events correspond to navigation actions such as a click on a button and target events to actually used features such as games or radio. Thus, a small number of transition events and a large number of target events corresponds to a professional usage of Anne. Italy shows an ideal course of these curves. The number of transition events is large at the beginning and reduces quickly over time whereas the number of target events has the inverse course. This shows concisely the learning effects regarding Anne over time. In the Netherlands, the number of target events approximately remained constant over time. The transition events inversely parabolically decreased. These users learned to navigate more efficiently but still did not use Anne more despite the resulting time gain. In Luxembourg, transition and target events are reduced similarly over time. This can be interpreted as a general reduction of usage without a sense of confidence. Overall, the telemetry analysis shows a consistent image of the countries. Italy had the largest usage activity and learning effects of Anne. Netherlands had less activity and learning effect followed by Luxembourg with an even more downward trend in these areas.

About usability data, Anne received a mean score of 66.2 (SD 19) that is slightly below the average score of 68 (SD 12.5). Minor differences among countries are reported in Table 2 as well as the analysis of the single items. Participants evaluated Anne as easy-to-use and quickly learnable, they felt a sense of confidence while using the system and expressed the willingness to use it frequently.

### 3.2. Closeness Scale and Role of Anne

The enrolled individuals visualized their relationship with Anne with no connection (16/50) or little connections (15/50), or some connections (10/50). Only four participants perceived equal or even strong relationships in three cases.

From the analysis of the open questions, 21 older adults perceived Anne as an assistant, companion, and reminder (Box 3), whereas the other 30 perceived the ECA as a distractor or did not attribute any role (Box 4).

Box 3Examples of positive quotations about Anne as an assistant, companion, and reminder.
*It is nice to have* Anne *in my room and to play the card game. I also have a paper agenda. I write everything down. I like that Anne speaks out about everything**It makes me more* confident *about myself when Anne says what to do, and I can read it in my agenda*
*I have a new buddy, that’s nice!*
*No panic anymore,* confident *that I will not miss medication or appointments*


Box 4Examples of negative quotations about Anne as a distractor or without a role.
She does not correspond to my needs except the gamesShe irritated me. She was constantly there and it made me aware that I am getting olderI don’t like using technology. I didn’t find Anna interestingI have other APPs in my tablet that perform more and better. I do not need it now as it is but maybe in the future.


### 3.3. Cluster Analysis and Personas to Identify Technology-Based Intervention in the Care Pathway

As reported in Table 3 and Figure 5, the cluster analysis identified two main groups that diverge for age class (*p* = 0.000), MMSE score (*p* = 0.020) and previous experience with tablet (*p* = 0.004): Cluster 1 and Cluster 2. 

Analyzing and comparing group characteristics, it emerged that in Cluster 1 subjects were aged 71 or more, their MMSE score was 24–27 in 76.2% vs. 43.3%, they have no experience in using technological devices (81.0% vs. 40.0% never used a tablet). This led to lower SUS scores (60.5, SD 24.8 vs. 70.9, SD 12.8), which can be considered nearly significant) and a higher proportion of subjects that thought that Anne had no role (38.1% vs. 11.1%). On the contrary, in Cluster 2, subjects aged less than 70, with a higher MMSE, scored 28–30 and had previous experience in using technologies. Within this group, the SUS score was good and a good proportion of subject that considered Anne as a friend or assistant to which they can feel close and who remind them of important things like appointments or when to take medication.

Since cluster analysis is often used to explore clusters of people within datasets for developing personas [27], two personas were generated (Figure 6). Each persona identifies patient profiles with different personal, socio-cultural, health and home environments, needs, and potential benefits derived from digital resources.

## 4. Discussion

A sample of 55 older adults previously diagnosed with dementia was involved in four weeks of using the embodied conversational agent called Anne in their home environment. From the analysis of the usability test, Anne was considered as easy-to-use and quickly learnable. Moreover, data gathered by the closeness scale showed that 21 participants perceived the virtual agent as a friend and assistant while 30 perceived the ECA as a distractor or did not attribute any role. The telemetry report detected different activity rankings in the three sites: at the beginning of the 4 weeks, users faced an exploratory period, trying out the many features and not knowing how to handle the device very well. Then, users’ actions become more efficient and purposeful. These cross-national findings endorse the optimism recently emerged about ECAs as promising tools to cope with the health and well-being of people with dementia. Despite this endorsement, the study also underlines the challenge of designing, developing, and assessing such human-like technology following a user-driven approach. The first challenge is to target the disease process from its earliest stages and follow the person throughout the journey to foster healthy aging and improve the lives of older people, their families, and the whole community. However, a further collection of data sources will be necessary to have a clear framework of triggers and barriers. Nevertheless, the cluster analysis showed different user characteristics that could be plausible to justify these results: Persona A and B matched different needs and underlined how technology-based interventions require a significant effort, personalized features, and patient-centered care pathways, to be effective for the different types of users.

Effectively, through the cluster analysis and the related Persona A and B developed in this study, some possible care pathways to promote technology-based interventions for people with dementia can be discussed.

Because of the condition’s progressive nature, people with dementia have complex problems with symptoms in many domains as well as complex needs [28]. According to the WHO [29,30], the signs and symptoms linked to dementia can be classified into three stages. The early stage of dementia is often unnoticed since common symptoms include forgetfulness, losing track of time, and becoming lost in familiar places, and they may be dismissed as part of normal aging. As dementia progresses to the middle stage, clearer and more restricting symptoms include becoming forgetful of recent events and people’s names, becoming lost at home, increasing difficulty with communication, needing help with personal care, and experiencing behavior changes, including wandering and repeated questioning. At the late stage of dementia, the person is almost dependent and inactive with serious memory instabilities such as becoming unaware of the time and place, having difficulty recognizing relatives and friends, having an increasing need for assisted self-care, having difficulty walking, experiencing behavior changes that may escalate and include aggression.

As aforementioned, in the majority of the cases, the main problem presented at first contact with the health care system is related to memory issues [31,32] and, especially, at this first stage, individuals are experiencing uncertainty, confusion, anxiety, worrying about memory problems, and future life [33]. Since the definition of clinical care pathways for dementia starts at this point, it could be strategic to introduce technologies at this early stage to guarantee familiarity [8] with new devices and assure the acquisition or improvement of digital competencies since perceived difficulties become more pronounced as dementia severity increases [34,35,36]. For example, the telemetry data reported in this study underlined how a period of familiarization is necessary to switch from an exploratory usage to an advanced one. The role of technologies at this stage is almost focused on the preservation or maintenance of residual capacities, all the individual’s physical and mental capacities, and its interactions with the relevant environmental characteristics that determine the functional ability of that person, which is central for healthy aging [37,38]. Indeed, as reported in Persona B, the promotion of knowledge about risk factors and potential prevention or detection methods, as well as cognitive stimulation and training, may have a protective effect on people with or without a genetic risk. This can be easily supported by technology-based interventions in the form of digital coaching. In this specific case, a person with digital literacy and the ability to use devices autonomously can meet their own needs when interacting with such systems but may ask for more personalization. For example, as reported in Box 4, the difficulty in matching with felt needs in that period of life or the comparison of other devices seen as more suitable in terms of services could be barriers to the adoption. Overcoming such barriers in this cluster of users could be tackled by supporting higher-level needs such as belonging, self-esteem, identity, and self-actualization [39] as is carried out in a person-centered approach which tailors care to the individual’s interests, abilities, history, and personality [28,40,41,42]. Moreover, the intervention must not be prescriptive but rather stimulate self-motivation.

On the contrary, in the middle stage of dementia, such as Persona A, technology-based interventions must move from preventative support to offering actual assistance and monitoring services. Functionalities such as planning and organizing day appointments or support with everyday tasks such as cooking, using the toilet, and dressing are all helpful in the maintenance of independent living in the home environment. In this case, a person without digital literacy could perceive the embodied conversation agent as more effective than other non-human-like devices through meeting the need for companionship and self-efficacy, as reported in the data shown in the results section.

Moreover, the interaction modality through the voice is the most natural and easy way to communicate, especially for people with dementia in the middle stage [8]. Despite this advantage, personalization remains a critical challenge. The system needs to be adapted to the felt needs, in accordance with a person-centered approach and linked also to the cultural, educational differences including the national/local care infrastructures (e.g., if people are mainly cared for in institutions or in their own homes; whether homes have access to fast internet, etc.). In addition, people with no or low device skills will require the help and support of family carers to initiate and master the initial interactions with the new devices. By providing opportunities for leisure and social activities thus facilitating participation and social inclusion, the embodied conversational agent can potentially help maintain physical and mental residual capacities. Indeed, the risk of loneliness and social isolation has been recognized as evident during the COVID-19 pandemic [43] with separation, loss, confusion, despair, and abandonment; stress and exhaustion exacerbation were detected as the main themes among people with dementia [44,45,46]. With regards to the late stage of dementia, for safety reasons, formal/professional caregivers may have to use the embodied conversational agent within a supervised environment (i.e., in nursing homes or daycare centers) because dramatic swings in mood and behavior can be frequent, and it is not predictable how individuals react in front of an unknown virtual assistant [19]. From a technical point of view, in the future, personalization may be completely individually tailored for each user or even more to their current status at the moment of interaction [47]. In this setup, telemetry data is used to give a dynamic response to the user’s status quo [48]. For example, Stricker et al. [49] develop methods to estimate the user’s attention and emotion from image sequences. Using these methods, the embodied conversational agent could react to negative user’s emotions and provide assistance. Wargnier et al. [37] developed an ECA that responds to user’s attentional disorders by calling the user to regain their attention. Benkaouar et al. [21] used a multisensory environment to detect when the user would like to interact with a device. Moreover, technological advances in the embodiment, content, communication modality, and strategy are needed as well as in-depth knowledge on preferences regarding the appearance, animation, and personalized features that can influence user acceptance and efficacy of the intervention. Indeed, further research with larger sample sizes, a significant length of trials and the collection of both qualitative and quantitative measures will be useful to go beyond the actual state of the art. Furthermore, the challenge to promote health and wellbeing for people living with dementia through human-centered technologies requires the partnership of research networks, medical scientists, technology developers, patients, and their formal and informal caregivers. Indeed, this study underlined how the multidisciplinary approach is the added value in the development of digital technologies. This is particularly needed for people with dementia as a heterogenic age group depending on personal diagnosis and dementia journey.

### Comparison with Prior Works and Limitations

Design for dementia is a complex area, and this poses several challenges when conducting research [45]. To our best knowledge, apart from Stara et al. [19], no other study within a similar setting such as the Living Well four-week use of an embodied conversation agent in a home setting among 55 individuals living with dementia and the usage of telemetry data, could be found in the current research literature.

Technically collecting user’s behavior data and analyzing it, as performed in natural sciences, gives a complementary view to understanding user’s behaviors.

Compared to data collected via questionnaires, a data-driven quantitative approach such as telemetry is not mediated through survey participants’ answers but through data objectively describing their behavior. In this context, interpreting the results is less evident and requires a qualitative understanding of the domain behind the data [46]. Furthermore, further challenges are faced, such as building an IT infrastructure that reliably collects the data and pre-processes it meaningfully while detecting instances of minor data quality. Moreover, this study explores how personas representing people with dementia in the early and middle stages, can be applied to the design of embodied conversational agents in the care pathway.

Despite these strengths, the study has some boundary conditions: a sample of 55 individuals coming from three different nations could be seen as too small to generalize findings. Indeed, a large population over a significant observational period would be preferable, as this would improve the significance of the statements deduced from the data.

The personas developed in this study were not reviewed and validated by participants or representatives of people with dementia, other than the clinical staff involved in the Living Well project. Such a review with the real users might be a further step to revise and better match the description of needs, barriers, and services.

## 5. Conclusions

As a result of social and physical distancing to prevent the spread of the COVID-19 pandemic, many people quickly adapted and transitioned to a virtual setting for continuity of care. This transition poses unprecedented challenges that are transforming health and care provision worldwide. Using data collected after a four-week usage of the embodied conversational agent called Anne in a home environment by 55 individuals living with dementia, this study (1) identified which socio-demographic characteristics are independently associated with the use of the embodied conversational agent, called Anne, among subjects with dementia after four weeks of usage in their home environments, (2) uncovered patient cluster profiles based on these characteristics, and (3) discuss technology-based interventions challenges. Therefore, this study enriches the open discussion on how innovative technologies contribute to the future of dementia care and, could guide future researchers and developers in investigating modalities for personalizing technology-based interventions that match the user’s characteristics and needs.

As the number of people with dementia is expected to grow and the lesson learned from the pandemic underlined the prominent role of technologies in mitigating social distancing and home-based care for those more vulnerable, we are reaching a stage where choosing to integrate technology alongside human-provided care is mandatory.

Although there are opportunities to advance the state of the art on dementia-focused embodied conversational agents in general, it is important that such virtual agents should be evidence based and designed by multidisciplinary teams, following patient-centered care as well as user-centered design (UCD) approaches [19].

## Figures and Tables

**Figure 1 ijerph-18-10405-f001:**
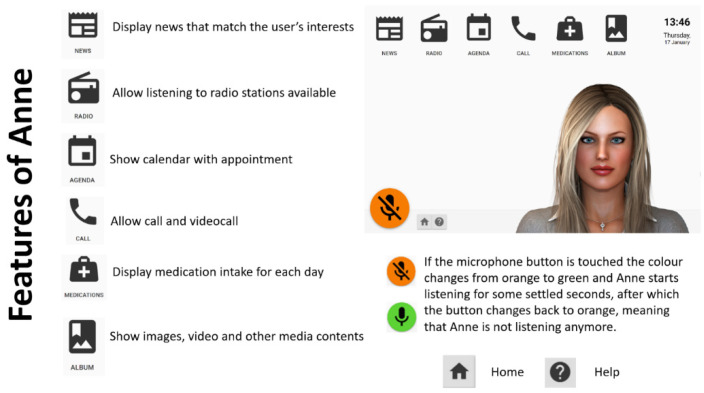
The ECA Anne.

**Figure 2 ijerph-18-10405-f002:**
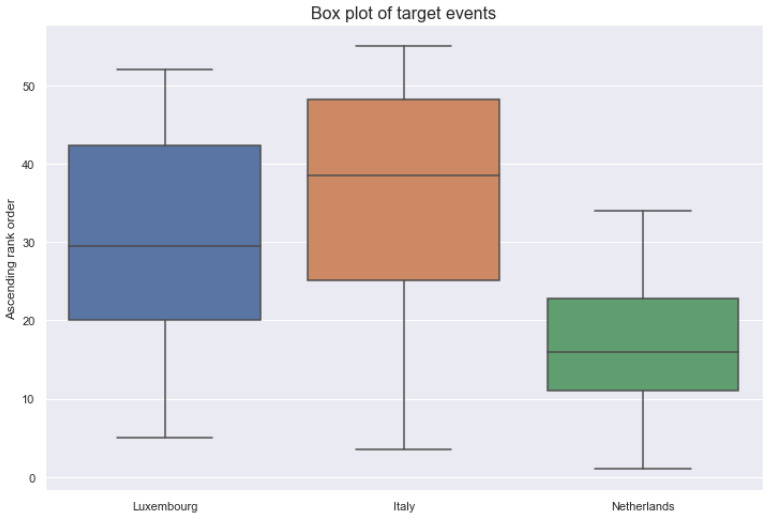
Box plot of target events for each country.

**Figure 3 ijerph-18-10405-f003:**
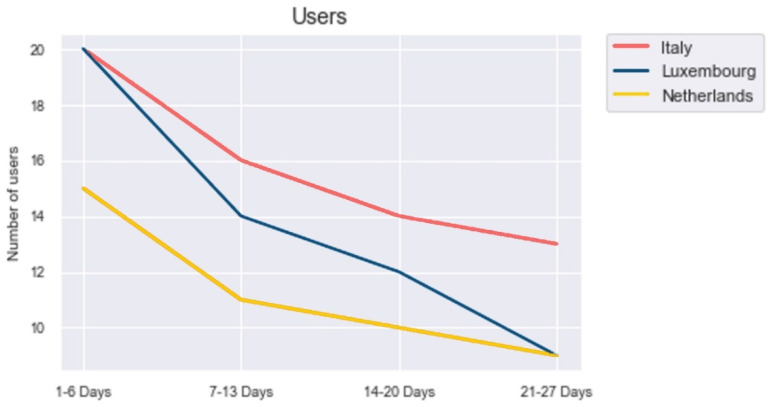
Number of users over time.

**Figure 4 ijerph-18-10405-f004:**
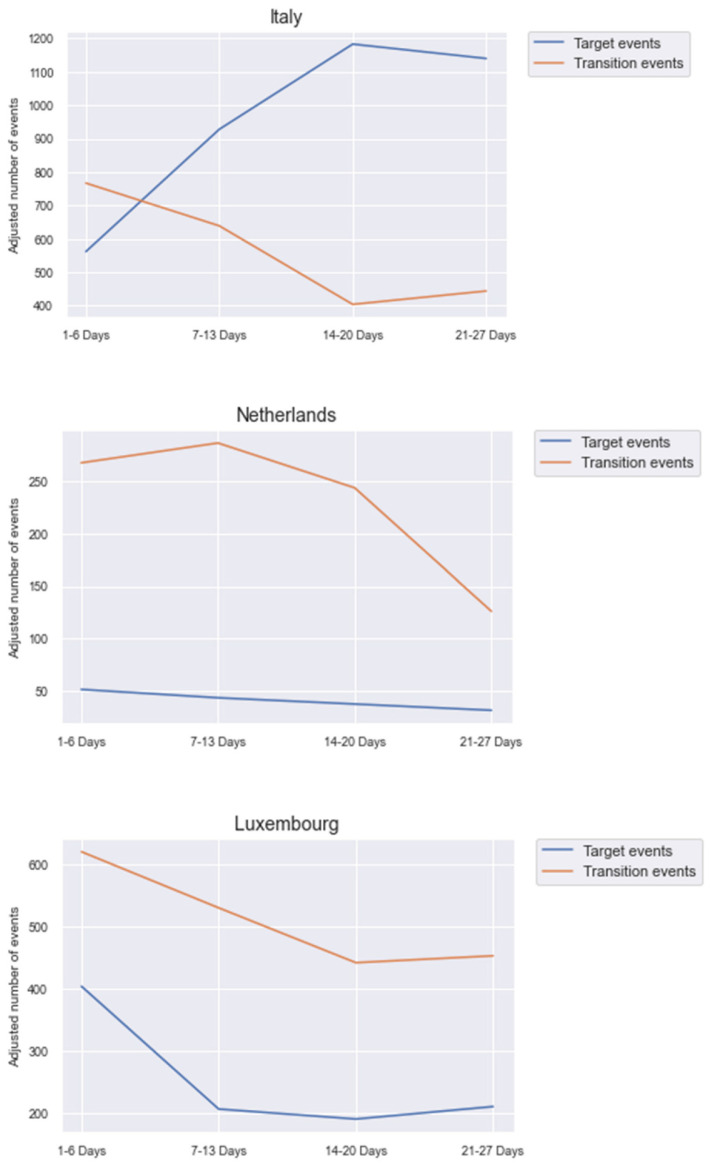
Transition and target events over time in the different countries (out of comparison reasons between the periods, the number of transition and target events were divided by the number of users in this period).

**Figure 5 ijerph-18-10405-f005:**
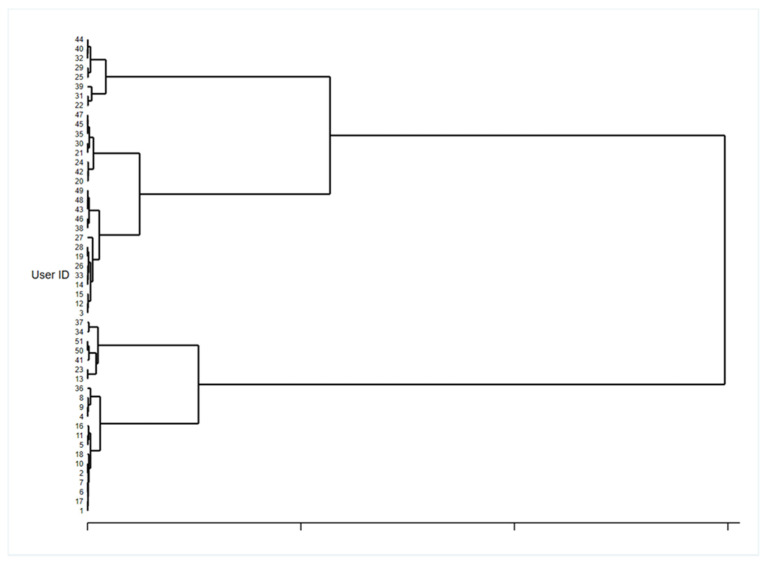
Hierarchical clustering dendrogram.

**Figure 6 ijerph-18-10405-f006:**
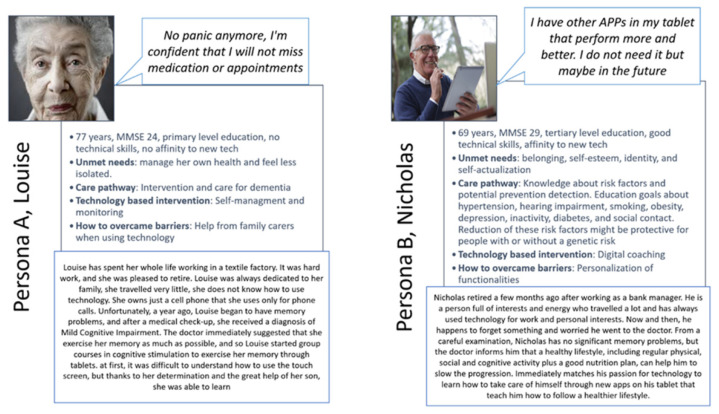
Persona A and B.

**Table 1 ijerph-18-10405-t001:** Socio-demographic data of participants.

	IT N = 20	LUX N = 20	NL N = 15	TOTAL N = 55
**Age, mean (SD)**	75.5 (4.2)	71.5 (11.2)	63.6 (9.7)	70.8 (9.9)
**Gender (%)**				
Male	30.0	45.0	33.3	36.4
Female	70.0	55.0	66.7	63.6
**Marital status (%)**				
Married	85.0	35.0	13.3	47.3
Full time relationship	0.0	0.0	0.0	0.0
Separated	0.0	5.0	0.0	1.8
Divorced	5.0	20.0	0.0	9.1
Single	5.0	15.0	86.7	30.9
Widowed	5.0	25.0	0.0	10.9
**Education (%)**				
No education	0.0	0.0	13.3	3.6
Primary	35.0	20.0	53.3	34.6
Secondary	20.0	70.0	33.4	41.8
Tertiary	45.0	10.0	0.0	20.0
**MMSE, mean (SD)**	25.2 (1.3)	29.2 (1.2)	26.6 (2.0)	27.0 (2.3)
**QoL pre, mean (SD)**	28.5 (6.6)	35.9 (6.3)	37.4 (3.7)	28.5 (6.6)
**QoL post, mean (SD)**	28.9 (7.8)	35.6 (4.8)	38.0 (2.3)	28.9 (7.8)
**delta Qol, mean (SD)**	0.4 (4.6)	−0.3 (5.3)	0.8 (1.6)	0.4 (4.6)

**Table 2 ijerph-18-10405-t002:** System Usability Scale average scores among participants.

SUS Items	IT	LUX	NL	TOT
Item 1. I think that I would like to use this system frequently	3.8 (1.3)	2.5 (1.4)	3.5 (0.8)	3.2 (1.3)
Item 2. I found the system unnecessary complex	1.8 (1.0)	1.7 (1.4)	1.8 1.0)	1.7 (1.1)
Item 3. I thought the system was easy to use	4.1 (1.2)	3.8 (1.4)	3.1 (1.2)	3.8 (1.3)
Item 4. I think that I would need the support of a technical person	2.9 (1.5)	2.2 (1.2)	2.2 (1.2)	2.2 (1.4)
Item 5. I found the various functions well integrated	3.7 (0.9)	3.1 (1.2)	2.5 (0.5)	3.1 (1.0)
Item 6. I thought there was too much inconsistency	2.4 (1.3)	2.6 (1.2)	2.3 (1.1)	2.6 (1.2)
Item 7. I would imagine that most people would learn quickly	3.9 (0.9)	3.7 (1.0)	3.3 (0.8)	3.7 (0.9)
Item 8. I found the system very cumbersome	2.0 (1.1)	2.2 (1.3)	1.7 (0.9)	2.2 (1.2)
Item 9. I felt very confident using the system	3.3 (1.5)	3.5 (1.2)	3.4 (1.0)	3.5 (1.3)
Item 10. I needed to learn a lot of things before I could get going	2.9 (1.3)	2.4 (1.3)	2.8 (1.0)	2.4 (1.4)
**SUS Score**	67.1 (23.3)	65.3 (16.7)	66.3 (15.2)	66.2 (19.0)

**Table 3 ijerph-18-10405-t003:** Cluster characteristics.

	Total	Cluster 1	Cluster 2	*p*-Value
	N = 51 (4 Missing)	N = 21	N = 30	
**Cluster analysis variables**				
**Gender, n (%)**				0.917
Female	32 (62.7)	13 (61.9)	19 (63.3)	
Male	19 (37.3)	8 (38.1)	11 (36.7)	
**Age class, n (%)**				0.000
<70	23 (45.1)	0 (0.0)	23 (76.7)	
71+	28 (54.9)	21 (100.0)	7(23.3)	
**Education, n (%)**				0.813
Low	20 (39.2)	8 (38.1)	12 (40.0)	
Medium	21 (41.2)	8 (38.1)	13 (43.3)	
High	10 (19.6)	5 (23.8)	5 (16.7)	
**MMSE score, n (%)**				0.020
24–27	29 (56.9)	16 (76.2)	13 (43.3)	
28–30	22 (43.1)	5 (23.8)	17 (56.7)	
**Have you ever used a tablet** **before this experience? n (%)**				0.004
Yes	22 (43.1)	4 (19.0)	18 (60.0)	
No	29 (56.9)	17 (81.0)	12 (40.0)	
**Outcome variables**				
**SUS score, mean (SD)**	66.22 (19.6)	60.48 (24.8)	70.87 (12.8)	0.070
**Closeness scale, mean (SD)**	2.40 (1.4)	2.14 (1.3)	2.62 (1.6)	0.273
**Role of Anne, n (%)**				0.027
Friend, Assistant, Reminder	37 (77.1)	13 (61.9)	24 (88.9)	
No Role	11 (22.9)	8 (38.1)	3 (11.1)	

## Data Availability

The data presented in this study are available on request from the corresponding author. The data are not publicly available due to privacy restriction.

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
