# Peer review of "Toward the Integration of Technology-Based Interventions in the Care Pathway for People with Dementia: A Cross-National Study"

_ijerph, 2021, doi:10.3390/ijerph181910405_

Round 1

Reviewer 1 Report

The article presents an interesting study in relation to the challenges that demographic change and ageing are driving innovation to find solutions and transform the delivery of health and care services worldwide.
The objectives of the study are: 1) identify which socio-demographic characteristics are independently associated with the use of the Embodied Conversational Agent among subjects with dementia, 2) un cover patient cluster profiles based on these characteristics, and 3) propose two personas to discuss technology-based interventions challenges.
This study is part of a European-wide study.
I enjoyed reading your article, and it is relevant to the mission of the journal, which means that the topic is well within the scope of the journal. The topic of the article is interesting and a timely study as the ageing of the population is a reality. Research is needed to facilitate the understanding of technology-based interventions in the care pathway for people with dementia. This will undoubtedly contribute to facilitating quality care for this group of people. Therefore, the study enriches the open discussion on how innovative technologies contribute to the future of Mentia care development.
In my opinion, it is a document that effectively contributes to the clarification of technology-based interventions for people with dementia.
The paper is well structured, which makes the study easy to understand.  It also has a good theoretical basis.
Aim: The research problem and the aim of the study are well defined.
Method: The study presents a four-week virtual agent study of 55 people with dementia living in their home environment.
The phases of the research are presented in a clear and structured way. 
Results: This evaluator considers that the results shown in terms of the study problem are relevant and lead to clear conclusions. The figures and tables presented contribute to the reader's understanding. 
The "Discussion" section is clear and correlates with other studies in the same direction.
I consider that the authors should specify what the authors have learned from the findings that can guide the educational community towards improvements in this field. 

Author Response

Response 1: Thank you for the endorsement. We integrated these sentences at the end of the discussion: “Moreover, technological advances in the embodiment, content, communication modality and strategy are needed as well as in-depth knowledge on preferences regarding the appearance, animation, and personalized features that can influence user acceptance and efficacy of the intervention. Indeed, further research with larger sample sizes, control groups, a significant length of trials and the collection of both qualitative and quantitative measures will be useful to go beyond the actual state of the art. Furthermore, the challenge to promote health and wellbeing for people living with dementia through human-centred technologies requires the partnership of research networks, medical scientists, technology developers, patients, and their formal and informal caregivers.

Reviewer 2 Report

One question: What is the advantage of using artificial intelligence over human assistance? This question needs to be answered and explained in the text.

The system presented in the work was not compared with others, making it difficult to analyze whether the results are adequate, as there is no reference standard.  

Author Response

  1. One question: What is the advantage of using artificial intelligence over human assistance? This question needs to be answered and explained in the text.

Response 1: Thank you for this significant question. We clarified this point in the Introduction: “Indeed, artificial intelligence brings the benefit of 24x7 health monitoring. Based on optimization algorithms such technologies continually collect and analyze data in order to assess risk level and share behavioral or care recommendations. Furthermore, this remote digital assistance can increase, not replacing, the ability of healthcare professionals to better understand the day-to-day patterns and needs of the people they care for, thus facilitating the quality of life of people with dementia and their caregivers. ”

  1. The system presented in the work was not compared with others, making it difficult to analyze whether the results are adequate, as there is no reference standard. 

Response 2: Thank you for this consideration. We added the following text in the description of the system (2.1. Anne, the Embodied Conversational Agent): “In comparison with other ECAs designed for people with dementia, this virtual agent represents a step forward: 1) Anne is based on a mobile standalone solution, whereas the other examples of virtual agents are displayed on standard television sets and computer screens; 2) Anne offers a multipurpose tool integrating features such as reminders (personal and medication agenda), communication (video calls), information (news), and entertainment (games and music) that support users in all aspects of daily life engaging them in various activities (i.e. help them pass the time in a more meaningful way and improve their quality of life). On the contrary, other ECA functions sought mainly to overcome memory problems, guide patients in their daily activities, and meet their need for communication and social interaction [8].”

Reviewer 3 Report

Reviewer’s Comments on ijerph-1346254

Toward the Integration of Technology-Based Interventions in 2 the Care Pathway for People with Dementia: A Cross-National 3 Study

This paper presents results of a study that considers the use of virtual agent for people with dementia, the results of the study are discussed, and some insights are provided. I have the following comments:

  1. I consider that the abstract could be complemented to give a clear idea of the reported research. The first sentence may be rearranged to make a smooth transition to the rest of the abstract.

  1. Discussion in section 3.3 may be extended, the authors do not mention Table 3 in the discussion of the cluster analysis. Please discuss the p value in table 3. Furthermore, please mention the considered dissimilarity method and the general method. Are the authors considering K-means algorithm?

  1. I consider that a graphic that denotes the different datapoints aligned to the two clusters should be included.

Minor comments:

  1. Although the term MMSE is presented (Mini Mental State Examination), it should be presented as “Mini Mental State Examination (MMSE)” the first time it is mentioned.

  1. Missing parenthesis in table 2.

Author Response

This paper presents results of a study that considers the use of virtual agent for people with dementia, the results of the study are discussed, and some insights are provided. I have the following comments:

  1. I consider that the abstract could be complemented to give a clear idea of the reported research. The first sentence may be rearranged to make a smooth transition to the rest of the abstract.

Response 1: thank you for this suggestion. We agree that the first sentence was not clear. We modified the abstract accordingly.

  1. Discussion in section 3.3 may be extended, the authors do not mention Table 3 in the discussion of the cluster analysis. Please discuss the p value in table 3. Furthermore, please mention the considered dissimilarity method and the general method. Are the authors considering K-means algorithm?

Response 2. Thank you for all these remarks. Table 3 and p values are now mentioned in the discussion. The k-means algorithm is used when non-hierarchical clustering is applied. In our manuscript we opted for agglomerative hierarchical clustering. The selected cluster fusion procedure was the Ward’s one. It was mentioned in the “statistical analysis” section, but now we described more extensively in order to avoid misunderstandings or misinterpretations

  1. I consider that a graphic that denotes the different datapoints aligned to the two clusters should be included.

Response 3. We thank the reviewer for this comment. In order to make clearer which independent variables were selected for the cluster analysis and which were the independent variables, we specified them in table 3. In this way main characteristics of the two clusters were more directly addressed, avoiding producing additional tables/figures since the manuscript is already plenty of them. In this way, we are able to satisfy all revisions requested by each reviewer.

Minor comments:

  1. Although the term MMSE is presented (Mini Mental State Examination), it should be presented as “Mini Mental State Examination (MMSE)” the first time it is mentioned.
  2. Missing parenthesis in table 2.

Responses 5 and 6: Thank you for noting these mistakes. We corrected the text.

Reviewer 4 Report

This paper investigates the socio-biographical characteristics associated with perceived ease of use and intention to use of Embodied Conversational Agents by patients with dementia. 

Overall, I find the paper concise and clear (more so in the introduction than in the discussion), and while cluster analysis is not that common, it seems like a good application here. 

However, I believe some key analyses are missing.

First of all, from the abstract I expected some kind of exploration - e.g., a linear model - of how socio-biographical characteristics related to SUS scores and closeness judgements. Currently, the paper presents some descriptive statistics and then skips to the cluster analysis, which doesn't really tell us how age, gender, MMSE scores, etc. relate with SUS scores/closeness. 

Secondly, it's strange to see a study that used telemetry to collect user behaviour over 4 weeks and then didn't explore how behaviour changed over that time. Did the 'typical' user increase or decrease usage over time? Are the clusters related to behaviour patterns?

It's also currently unclear which variables were used to identify the clusters. 

On the other hand, extensive space is given to presenting and discussing personas, which feel out of place in a scientific paper. I believe shifting the focus of the paper away from personas and towards telemetry data would give more solid foundation to the discussion and conclusions. 

Author Response

This paper investigates the socio-biographical characteristics associated with perceived ease of use and intention to use of Embodied Conversational Agents by patients with dementia. 

Overall, I find the paper concise and clear (more so in the introduction than in the discussion), and while cluster analysis is not that common, it seems like a good application here. 

However, I believe some key analyses are missing.

1 First of all, from the abstract I expected some kind of exploration - e.g., a linear model - of how socio-biographical characteristics related to SUS scores and closeness judgements. Currently, the paper presents some descriptive statistics and then skips to the cluster analysis, which doesn't really tell us how age, gender, MMSE scores, etc. relate with SUS scores/closeness. 

Response 1. Thank you for this comment. We checked for statistically significant relationships in bivariate and linear regression but there were not. In any case our aim was to find possible users’ profiles therefore we merged the characteristics of the subjects via agglomerative cluster analysis.

2 Secondly, it's strange to see a study that used telemetry to collect user behaviour over 4 weeks and then didn't explore how behaviour changed over that time. Did the 'typical' user increase or decrease usage over time? Are the clusters related to behaviour patterns?

Response 2. Thank you for underlining this missing part. We added the behaviour changed over time (both graphs and content). The telemetry collected anonymous data so we cannot related clusters and telemetry patterns due to legal and ethical constraints.

3 It's also currently unclear which variables were used to identify the clusters. 

Response 3. We thank the reviewer for this comment. In order to make clearer which independent variables were selected for the cluster analysis and which were the independent variables, we specified it in table 3. In this way main characteristics of the two clusters were more directly addressed, avoiding to produce additional tables/figures since the manuscript is already plenty of them.

4 On the other hand, extensive space is given to presenting and discussing personas, which feel out of place in a scientific paper. I believe shifting the focus of the paper away from personas and towards telemetry data would give more solid foundation to the discussion and conclusions. 

Response 4. Thank you for this suggestion. We changed the discussion and deleted the personas from the elicitation of our objectives, trying to adhere to each reviewer's different vision and the aim of the special issue that cited the Person-centred care and Kitwood’s concept. We initially pushed the Personas as a way to match the personhood demands. We hope that this revision suits your standards.

Round 2

Reviewer 2 Report

The review improved the reviewer's understanding. With the additional documents made available, some doubts were resolved. This reviewer feels contemplated, although he recommends improving the quality of the figures.

Author Response

Thank you so much for your time and effort. We will improve the quality of imagines during the editing process. Best regards

Reviewer 3 Report

The authors have considered the provided comments to improve the paper, I have no further comments.

Author Response

Thank you for your time and effort. Best regards.

Reviewer 4 Report

The Authors met my previous requests satisfactorily. 

Author Response

Thank you again for your time and effort in supporting our manuscript. Best regards.